# DropPos: Pre-Training Vision Transformers by Reconstructing Dropped Positions

**Haochen Wang**[1,3]   **Junsong Fan**[1,4]   **Yuxi Wang**[1,4]   **Kaiyou Song**[2]
**Tong Wang**[2]   **Zhaoxiang Zhang**[1,3,4*]

[1]Center for Research on Intelligent Perception and Computing (CRIPAC),
State Key Laboratory of Multimodal Artificial Intelligence Systems (MAIS),
Institute of Automation, Chinese Academy of Sciences (CASIA)
[2]Megvii Technology    [3]University of Chinese Academy of Sciences (UCAS)
[4]Centre for Artificial Intelligence and Robotics, HKISI_CAS

{wanghaochen2022, junsong.fan, zhaoxiang.zhang}@ia.ac.cn
yuxiwang93@gmail.com   {songkaiyou, wangtong}@megvii.com

## Abstract

As it is empirically observed that Vision Transformers (ViTs) are quite insensitive to the order of input tokens, the need for an appropriate self-supervised pretext task that enhances the location awareness of ViTs is becoming evident. To address this, we present DropPos, a novel pretext task designed to *reconstruct **Dropped Positions***. The formulation of DropPos is simple: we first drop a large random subset of positional embeddings and then the model *classifies the actual position* for each non-overlapping patch among all possible positions *solely* based on their visual appearance. To avoid trivial solutions, we increase the difficulty of this task by keeping only a *subset* of patches visible. Additionally, considering there may be different patches with similar visual appearances, we propose position smoothing and attentive reconstruction strategies to relax this classification problem, since it is *not* necessary to reconstruct their *exact* positions in these cases. Empirical evaluations of DropPos show strong capabilities. DropPos outperforms supervised pre-training and achieves competitive results compared with state-of-the-art self-supervised alternatives on a wide range of downstream benchmarks. This suggests that explicitly encouraging spatial reasoning abilities, as DropPos does, indeed contributes to the improved location awareness of ViTs. The code is publicly available at https://github.com/Haochen-Wang409/DropPos.

## 1   Introduction

Learning extensible visual representations without any human annotations, known as self-supervised learning (SSL), has become a research hotspot in the field of computer vision [7,9,26,27,29,47,59,62] since it enables efficient transfer to various downstream benchmarks. To achieve this goal, researchers carefully design visual pretext tasks to produce appropriate supervision using *only* images [3, 18, 46, 52, 62, 69, 70]. Two popular approaches among these are contrastive learning (CL) [62] and masked image modeling (MIM) [3], both of which have demonstrated promising scaling behavior of vision models, particularly Vision Transformers (ViTs) [21]. Despite the success of these methods, ViTs are found to be relatively *insensitive* to the order of input tokens [13, 43, 67], leading to the hypothesis that they tend to model the relationship between a set of *unordered* input tokens. Therefore, a natural question arises: beyond the current CL and MIM paradigms, *whether a pretext task that explicitly enhances the positional awareness of ViTs can further improve their representation learning abilities?*

---

[*]Correponding author.

37th Conference on Neural Information Processing Systems (NeurIPS 2023).

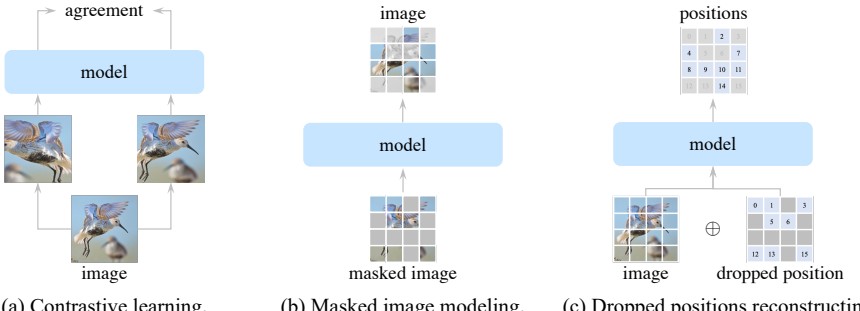

|  (a) Contrastive learning. | (b) Masked image modeling. | (c) Dropped positions reconstructing. |

Figure 1: **Comparison between different pretext tasks. (a)** Contrastive learning aims to maximize the agreement between different views of one image. **(b)** Masked image modeling predicts specific contents of masked patches. **(c)** DropPos reconstructs positions *solely* based on visual appearance.

To answer this question, we begin by revisiting the forward procedure of ViTs. A sequence of positional embeddings (PEs) [51] is added to patch embeddings to preserve position information. Intuitively, simply *discarding* these PEs and requesting the model to reconstruct the position for each patch naturally becomes a qualified location-aware pretext task. By forcing the model *only* to utilize visual appearance for position reconstruction, the model is supposed to learn the shape relationships and layouts of image contents, thereby improving the ability to encode spatial visual content. However, despite the simplicity and intuitiveness of this idea, position-related methods such as [18, 44] have fallen far behind. We identify several difficulties in designing this type of paradigm:

*(i)* Discarding all PEs brings *discrepancies between pre-training and fine-tuning* because the model has *never* been exposed to any PEs during pre-training. *(ii)* Strengths of ViTs in modeling long-range dependencies may cause them to solve the task superficially, leading to trivial solutions that *fail to learn highly semantic representations by solving this simple task. (iii)* Patches with similar visual appearances may result in confusing reconstruction targets, making it crucial to *decide which position to reconstruct precisely.*

Driven by this analysis, we present a novel, straightforward, and highly effective pretext task for self-supervised visual representation learning: *reconstructing **Dropped Positions** (DropPos). DropPos tackles the above issues systematically: *(i)* To address discrepancies, simply *dropping* a large subset of PEs instead of discarding all of them works effectively. *(ii)* To avoid trivial solutions, we only keep a *subset* of patches visible during pre-training, forcing ViTs to reconstruct the position of each visible patch with only *partial* inputs. *(iii)* To prevent ViTs from being overwhelmed by this particular task, we propose to relax this problem using position smoothing and attentive reconstruction strategies.

DropPos can be easily formulated into a simple patch-wise classification problem. Here is how it works: First, input patches are randomly masked and a *large* subset of PEs corresponding to these visible patches is dropped. Then, ViTs are tasked with classifying the positions of visible patches among all possible candidates, relying on *partial* observations and a few anchors (*i.e.*, patches with unmasked PEs). Fig. 1 illustrates two benefits of DropPos: *(i)* requiring less prior knowledge of data augmentation techniques compared to CL, and *(ii)* eliminating the need for a careful selection of target representation [39] and mask strategy [8, 54, 56] as MIM does.

We conducted extensive experiments to evaluate the performance of DropPos in comparison to state-of-the-art alternatives across various downstream tasks. With only 800 epochs of pre-training, DropPos achieves 84.2% top-1 accuracy on ImageNet-1K [48] validation set using ViT-B/16 [21], outperforming MAE [28] pre-trained with 1600 epochs by +0.6%. Moreover, on COCO [38] object detection/segmentation, and ADE20k [71] semantic segmentation benchmarks, which requires more abilities of spatial reasoning, DropPos surpasses MAE [28] *consistently*.

## 2   Related work

**Self-supervised learning.** The key challenge of self-supervised learning is the design of pretext tasks [18, 28, 32, 42, 44, 45, 50, 54, 69], *i.e.*, how to produce appropriate supervision signals using *only* images. Among them, contrastive learning [5, 9, 26, 29, 58, 60, 62, 66] has been popular recently, which seeks to maximize agreement between different views of the same images. Nevertheless, the

performance of contrastive-related methods highly depends on carefully-designed data augmentation techniques [5, 6, 9, 11, 12, 72]. Another mainstream is masked image modeling [1, 3, 10, 28, 54, 56], which involves predicting *specific contents* masked patches [2, 3, 19, 20, 28, 54, 64, 72]. Unlike these methods, DropPos offers a novel alternative for self-supervised ViTs: reconstructing dropped positions based on *only* visual clues. In this way, the model is urged to learn the spatial arrangement of local image patches, contributing to better localization abilities, which is important for spatial reasoning recognition tasks, *e.g.*, segmentation [22, 53, 55, 57, 58] and video understanding [17, 25].

**Position prediction in self-supervised learning.** Doersch *et al.* [18] first split an image into 3×3 grids and train a network to predict the *relative position* of paired patches from the same image. This problem is formulated as an 8-way classification. Noroozi and Favaro [44] extended this approach to solve "jigsaw puzzles" by urging the model to *predict the order* of *shuffled* non-overlapping crops. These approaches were originally developed using ConvNets [33] and thus lacked the ability to learn long-range dependencies, which is crucial for position prediction. Very recently, Zhai *et al.* [67] revisited this type of pretext task in the scope of Transformers [51] by pre-training ViTs [21] to predict patch positions *solely* based on their visual appearance, while discarding positional embeddings. However, these pre-trained models have *never* been exposed to positional embeddings, leading to discrepancies between pre-training and fine-tuning, and thus their performances are significantly inferior to state-of-the-art alternatives [6, 13, 28, 54]. Sameni *et al.* [49] come up with an auxiliary position-related objective and combine it with the popular contrastive learning paradigm. Caron *et al.* [4] extended this idea by predicting the relative location of a *query* (*i.e.*, local crop) to the corresponding *reference* (*i.e.*, global crop). However, [4] *only* examined the effectiveness on segmentation benchmarks [15, 24, 41, 71]. In contrast, DropPos incorporates a novel dropped position reconstruction objective and thus achieves competitive performances on a variety of downstream tasks, including classfication [48], detection [38], and segmentation [38, 71].

**Positional embeddings in Vision Transformers.** In the standard forward procedure of ViTs [21], learnable positional embeddings (PEs) are added with patch embeddings. Improving PEs and introducing more inductive bias into ViTs has become an active research area [23]. This is because some research shows that ViTs are surprisingly robust against permutations of the order of input patches. For instance, Chen *et al.* [13] demonstrate that the model performs well in classification even with *no* position embedding, suggesting that ViTs have *not* fully leveraged the positional information. Additionally, Naseer *et al.* [43] shows that ViTs are much *less susceptible* to patch shuffling perturbations than ConvNets. Therefore, we propose DropPos, a novel self-supervised pretext task that explicitly makes ViTs location-aware and promotes the emergence of spatial features.

## 3 Method

**Preliminary: Vision Transformers.** Given an image $\mathbf{I} \in \mathbb{R}^{H \times W \times C}$, it is first reshaped into a sequence of patches $\mathbf{I}_p \in \mathbb{R}^{N \times (P^2 C)}$, where $(H, W)$ indicates the spatial resolution, $C$ is the number of channels, $P$ is the patch size, and $N = HW/P^2$ is the number of patches. A linear projection is then applied to $\mathbf{I}_p$, mapping it to $D$ dimensions to get patch embeddings $\mathbf{x} \in \mathbb{R}^{N \times D}$. Also, a [CLS] token $\mathbf{x}_{\text{cls}} \in \mathbb{R}^D$ is used to aggregate the information. Position embeddings $\mathbf{p} \in \mathbb{R}^{(N+1) \times D}$ are added to the patch embeddings to retain positional information. Then, $\mathbf{z} = [\mathbf{x}_{\text{cls}}; \mathbf{x}] \oplus \mathbf{p}$ is the input of transformer blocks [51] which consists of a stack of multi-head self-attention mechanisms, where $\oplus$ denotes element-wise plus. In particular, $\mathbf{p}_0$ denotes the position embeddings of the [CLS] token, which will never be dropped.

### 3.1 DropPos

Fig. 2 illustrates our proposed DropPos. We first randomly generate a binary patch mask $\mathbf{M} \in \{0, 1\}^N$, where 1 means visible and 0 means masked, respectively. Note that the [CLS] token, *i.e.*, $\mathbf{z}_0$, is *always* visible in our setting, and thus the length of $\mathbf{M}$ is $N$ instead of $N + 1$ for simplicity. Given a pre-defined mask ratio $\gamma \in (0, 1)$, $\mathbf{M}$ is supposed to subject to

$$\sum_{i=0}^{N-1} \mathbf{M}_i = (1 - \gamma)N, \tag{1}$$

which means there are $\gamma N$ masked patches in total. It is worth noticing that we use patch masks to increase the difficulty of the pretext task, which is quite different from building a pretext task to

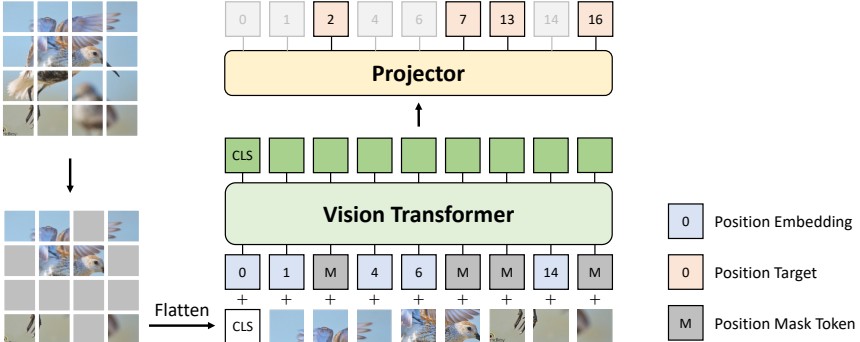

Figure 2: **Illustration of DropPos.** We first mask a large random subset of input images. Then, positional embeddings of visible patches are randomly dropped and [MASK] tokens are introduced. A lightweight projector is adopted to *reconstruct* those dropped positions. A simple patch-wise classification objective is adopted and gray position targets do not contribute to training.

recover input images in MIM [3, 28]. Then, we denote $\mathbf{x}_{\mathrm{vis}} = \mathtt{gather}(\mathbf{x}, \mathbf{M}) \in \mathbb{R}^{(1-\gamma)N \times D}$ to be the visible tokens, and position embeddings (PEs) for these unmasked tokens are randomly dropped, where $\mathtt{gather}(\cdot, \cdot)$ means only those unmasked patches (*i.e.*, $\mathbf{M}_i = 1$) are reserved. To address possible discrepancies, we aim to *drop* a subset of PEs of these visible patches instead of simply discarding all of them. Thus, we randomly generate a binary positional mask $\mathbf{M}_{\mathrm{pos}} \in \{0, 1\}^{(1-\gamma)N}$. Given a pre-defined position mask ratio $\gamma_{\mathrm{pos}} \in (0, 1)$, $\mathbf{M}_{\mathrm{pos}}$ is supposed to subject to

$$\sum_{i=0}^{(1-\gamma)N-1} \mathbf{M}_{\mathrm{pos}}^i = (1 - \gamma_{\mathrm{pos}})(1 - \gamma)N, \tag{2}$$

which indicates that the model is supposed to reconstruct $\gamma_{\mathrm{pos}}(1 - \gamma)N$ dropped positions based on remaining $(1 - \gamma_{\mathrm{pos}})(1 - \gamma)N$ anchor patches and the [CLS] token. PEs for visible patches $\mathbf{p}_{\mathrm{vis}} = \mathtt{gather}(\mathbf{p}_{1:N}, \mathbf{M})$ are gathered first. Then, we aim to obtain the final PEs $\mathbf{p}' \in \mathbb{R}^{(1-\gamma)N \times D}$ by replacing those dropped positions with a learnable [MASK] token:

$$\mathbf{p}_i' = \begin{cases} \mathbf{p}_0, & \text{if } i = 0, \\ \mathbf{p}_{\mathrm{vis}}^{i-1}, & \text{if } \mathbf{M}_{\mathrm{pos}}^{i-1} = 1, \\ \mathbf{p}_{\mathrm{mask}}, & \text{otherwise}, \end{cases} \tag{3}$$

where $i = 0, 1, \ldots, (1 - \gamma)N$ is the patch index, and $\mathbf{p}_{\mathrm{mask}} \in \mathbb{R}^D$ is the [MASK] token. Finally, the input of the Transformer encoder is $\mathbf{z}' = [\mathbf{x}_{\mathrm{cls}}; \mathbf{x}_{\mathrm{vis}}] \oplus \mathbf{p}' \in \mathbb{R}^{[(1-\gamma)N+1] \times D}$.

**Implementation.** Pre-training ViTs using our DropPos can be implemented efficiently and requires few specialized operations. The process involves three stages prior to forwarding features to the encoder. First, we randomly mask a subset of embedded patch tokens. Following MAE [28], the list of tokens is shuffled randomly, and the last portion of the list is removed. Next, positional embeddings of those unmasked patches are randomly dropped using the *same* masking operation. Then, positional mask tokens are introduced to obtain the final positional embeddings, which are added to the patch tokens finally and encoded

**Algorithm 1** Pseudo-Code of DropPos.

```
# x: embedded patch tokens
# gamma/gamma_pos: patch/position mask ratio
# mask_token: learnable [MASK] token

# patch masking, [CLS] token is kept visible
ids_vis, _ = masking(x, gamma)
x_vis = gather(x, ids_vis)
# position dropping for visible patches
pos_embed = gather(pos_embed, ids_vis)
ids_keep, ids_res = masking(pos_embed, gamma_pos)
pos_embed = gather(pos_embed, ids_keep)
# concat with mask tokens
pos_embed = cat([pos_embed, mask_token])
# restore
pos_embed = gather(pos_embed, ids_res)
# add to patch tokens for forward pass
return x_vis + pos_embed
```

by ViTs followed by a lightweight decoder. A simple classification objective is applied to those unmasked patches introduced as follows.

## 3.2 Pre-training DropPos

DropPos is pre-trained by reconstructing dropped positions. Concretely, as illustrated in Fig. 2 and presented in Sec. 3.1, an ViT [21] $f_\theta$ parameterized by $\theta$, takes $\mathbf{z}'$ as input. A lightweight decoder $g_\psi$ parameterized by $\psi$ then projects $f_\theta(\mathbf{z}')$ to patch-wise position predicitons $\mathbf{o} = g_\psi(f_\theta(\mathbf{z}')) \in \mathbb{R}^{(1-\gamma)N \times N}$. We omit the [CLS] token here for simplicity. For each *visible* patch $i \in \{0, 1, \dots, (1 - \gamma)N - 1\}$, $\mathtt{softmax}(\mathbf{o}_i) \in \mathbb{R}^N$ represents the probability density function of the predicted positions. Next, we have to produce appropriate ground truths, *i.e.*, the actual position for each visible patch. To this end, we gather positions via $\mathbf{y} = \mathtt{gather}([0, 1, \dots, N-1], \mathbf{M}) \in \mathbb{R}^{(1-\gamma)N}$, where vector $[0, 1, \dots, N-1]$ is naturally the actual position for all patches before masking (w/o the [CLS] token). Overall, the objective is simply the vanilla cross-entropy loss given true positions $\mathbf{y}_{ij}$:

$$\mathcal{L} = - \sum_{i=0}^{(1-\gamma)N-1} \sum_{j=0}^{N-1} (1 - \mathbf{M}_{\mathrm{pos}}^i) \cdot \mathtt{one\_hot}(\mathbf{y}_{ij}) \cdot \log \left[ \frac{\exp(\mathbf{o}_{ij})}{\sum_{k=0}^{(1-\gamma)N-1} \exp(\mathbf{o}_{ik})} \right]. \tag{4}$$

*Only* patches with dropped positions, *i.e.*, orange patches in Fig. 2, contribute to this objective, which is reflected by the term $(1 - \mathbf{M}_{\mathrm{pos}}^i)$ in Eq. (4). Next, several techniques are proposed to prevent ViTs from being overwhelmed by this particular task, making them pay more attention to model spatial dependencies instead of simply reconstructing positions.

**Position smoothing.** Considering that different categories (*i.e.*, positions) are *not* completely independent under this setting, the classification problem in Eq. (4) is relaxed when the model has predictions *close* to the actual position. Specifically, a weight matrix $\mathbf{w} \in (0, 1)^{N \times N}$ is defined to measure similarities between different patches:

$$w(i, j) = \exp \left( -\frac{\mathrm{dist}(i, j)}{\sigma^2} \right), \tag{5}$$

where $\mathrm{dist}(i, j)$ means the Euclidean distance between patch $i$ and $j$, and $\sigma$ is a hyper-parameter. The matrix then should be normalized by $w^*(i, j) = w(i, j)/\left( \sum_{k=0}^{N-1} w(i, k) \right)$. Here, $\mathbf{w}_i^* \in \mathbb{R}^N$ indicates the smoothed ground truth when the actual position is $i$. Overall, the smoothed objective is:

$$\mathcal{L}_{\mathrm{smooth}} = - \sum_{i=0}^{(1-\gamma)N-1} \sum_{j=0}^{N-1} (1 - \mathbf{M}_{\mathrm{pos}}^i) \cdot w^*(\mathbf{y}_{ij}, j) \cdot \log \left[ \frac{\exp(\mathbf{o}_{ij})}{\sum_{k=0}^{(1-\gamma)N-1} \exp(\mathbf{o}_{ik})} \right]. \tag{6}$$

Furthermore, we relax the problem in Eq. (6) by setting a large $\sigma$ at the early stage, and we gradually *decay* the value of $\sigma$ to produce a challenging pretext task. A simple linearly decay is performed:

$$\sigma_t = \frac{t}{T}(\sigma_T - \sigma_0) + \sigma_0, \tag{7}$$

where $t$ and $T$ denote the iteration and the number of total steps, respectively. $\sigma_0$ and $\sigma_T$ are the initial value and final value of $\sigma$, respectively. We set $\sigma_0 = 1$ and $\sigma_T = 0$ by default.

**Attentive reconstruction.** Since there may be different patches that share similar visual appearance (*e.g.*, two blue patches that represent the sky), it is *not* necessary to reconstruct their *exact* positions. Simply swapping these patches still maintains reasonable visual coherence. Therefore, we leverage the feature similarities between the [CLS] token and patch tokens to be an extra weight of Eq. (6):

$$\mathcal{L}_{\mathrm{smooth}}^{\mathrm{attn}} = - \sum_{i=0}^{(1-\gamma)N-1} \sum_{j=0}^{N-1} (1 - \mathbf{M}_{\mathrm{pos}}^i) \cdot A_{\mathbf{y}_{ij}} \cdot w^*(\mathbf{y}_{ij}, j) \cdot \log \left[ \frac{\exp(\mathbf{o}_{ij})}{\sum_{k=0}^{(1-\gamma)N-1} \exp(\mathbf{o}_{ik})} \right], \tag{8}$$

where $\mathbf{A} \in \mathbb{R}^N$ denotes the similarity matrix. Concretely, $A_i$ means the affinity between the [CLS] token and patch token $i$. Let $\mathbf{f}_{\mathrm{cls}} \in \mathbb{R}^D$ and $\mathbf{f} \in \mathbb{R}^{N \times D}$ be the features output by the encoder $f_\theta$, and thus the affinity $A_i$ is computed by

$$\mathbf{A}_i = \frac{\exp(\cos(\mathbf{f}_{\mathrm{cls}}, \mathbf{f}_i)/\tau)}{\sum_{j=0}^{N-1} \exp(\cos(\mathbf{f}_{\mathrm{cls}}, \mathbf{f}_j)/\tau)}, \tag{9}$$

where $\tau$ indicates the temperature parameter and is set to 0.1 by default.

Table 1: **Ablation study of patch mask ratio** $\gamma$. Note that "$\gamma = 0$" means that the whole image is visible. We report the *averaged* top-1 accuracy of position predictions.

| $\gamma$ | ImageNet-1K | | COCO detection | | | COCO segmentation | | | ADE20k | |
|---|---|---|---|---|---|---|---|---|---|---|
| | fine-tune | position | $AP^b$ | $AP^b_{50}$ | $AP^b_{75}$ | $AP^m$ | $AP^m_{50}$ | $AP^m_{75}$ | mIoU | aAcc |
| 0.00 | 81.94 | 59.79 | 34.10 | 52.80 | 37.17 | 31.24 | 50.27 | 33.03 | 31.66 | 75.86 |
| 0.25 | 82.02 | 79.62 | 37.05 | 56.23 | 40.18 | 33.60 | 53.51 | 35.84 | 32.80 | 76.92 |
| 0.50 | 82.78 | 87.27 | 38.45 | 57.83 | 41.87 | 34.88 | 55.14 | 37.50 | 36.33 | 78.25 |
| 0.75 | **82.96** | 87.83 | **42.14** | 61.99 | 46.38 | **37.93** | 59.23 | 40.76 | **40.68** | 80.14 |
| 0.90 | 82.81 | 65.12 | 41.06 | 60.92 | 44.57 | 37.04 | 58.23 | 39.62 | 40.30 | 80.09 |

Table 2: **Ablation study of position patch mask ratio** $\gamma_{pos}$. Note that "$\gamma_{pos} = 1$" means that we do not provide any reference patch, *i.e.*, all visible patches are randomly shuffled. We report the *averaged* top-1 accuracy of position predictions.

| $\gamma_{pos}$ | ImageNet-1K | | COCO detection | | | COCO segmentation | | | ADE20k | |
|---|---|---|---|---|---|---|---|---|---|---|
| | fine-tune | position | $AP^b$ | $AP^b_{50}$ | $AP^b_{75}$ | $AP^m$ | $AP^m_{50}$ | $AP^m_{75}$ | mIoU | aAcc |
| 0.25 | 82.71 | 65.19 | 37.98 | 57.65 | 41.39 | 34.61 | 54.79 | 36.93 | 36.15 | 78.16 |
| 0.50 | 82.86 | 86.70 | 39.23 | 58.77 | 42.99 | 35.60 | 56.23 | 37.89 | 37.82 | 78.60 |
| 0.75 | **82.96** | 87.83 | **42.14** | 61.99 | 46.38 | **37.93** | 59.23 | 40.76 | **40.68** | 80.14 |
| 1.00 | 82.66 | 19.44 | 41.48 | 61.50 | 45.18 | 37.41 | 58.81 | 39.92 | 39.98 | 80.19 |

# 4 Experiments

**Pre-training.** We perform self-supervised pre-training on the ImageNet-1K [48] training set with a resolution of 224×224. By default, we take ViT-B/16 [72] as the backbone and perform 200 epochs of pre-training. The decoder is a stack of Transformer [51] and has a depth of 2 and a width of 512. Patch mask ratio $\gamma$ and position mask ratio $\gamma_{pos}$ are both 0.75 by default. Our implementation is based on HPM [54]. Details can be found in *Supplementary Material*.

**Evaluation.** We perform supervised training to evaluate our DropPos with end-to-end fine-tuning on ImageNet-1K [48] for classification. By default, 100 epochs of fine-tuning are performed following common practices [28, 54] for ViT-B/16 [21]. We report top-1 validation accuracy of a single 224×224 crop. As for COCO [38], following previous methods [28, 54], we take Mask R-CNN [30] with FPN [37] as the detector. We perform end-to-end fine-tuning on COCO [38] for 1× schedule with 1024×1024 resolution. We report $AP^b$ for object detection and $AP^m$ for instance segmentation, respectively. Our implementation is based on detectron2 [61] and ViTDet [36]. For ADE20k [71], following previous methods [28, 54], we take UperNet [63] as the decoder and perform end-to-end fine-tuning on ADE20k [71] with 512×512 resolution for 80k iterations. We take mIoU [24] as the main evaluation metric. Our implementation is based on mmsegmentation [14].

## 4.1 Ablation studies

In Tabs. 1 to 4, we take the ViT-B/16 [21] pre-trained with 200 epochs on ImageNet-1K [48] as the backbone. We highlight the default settings of our DropPos. Concretely, if not specified, the default setting is $\gamma = 0.75$, $\gamma_{pos} = 0.75$, $\sigma_0 = 1$, $\sigma_T = 0$, and $\tau = 0.1$. By default, 100 epochs of fine-tuning on ImageNet-1K [48], 1× schedule fine-tuning on COCO [38], and 80k iterations fine-tuning on ADE20k [71] are performed.

To evaluate the performance on the pretext task, *i.e.*, reconstructing dropped positions, we also report the *averaged* top-1 accuracy of position predictions on the ImageNet-1K *validation* set in Tabs. 1 to 4. We vary $\gamma \in \{0, 0.25, 0.5, 0.75\}$ and $\gamma_{pos} \in \{0.25, 0.5, 0.75, 0.95\}$ when measuring the position prediction accuracy, and average the position accuracy among 16 different cases.

**Main properties.** Illustrated by Tabs. 1 to 4, we find sufficient evidence to support the three difficulties claimed in Sec. 1. *(i)* In Tab. 1, *ViTs fail to learn highly semantic representations by simply solving the position reconstruction task*, because the performances of $\gamma = 0$ significantly lag behind. *(ii)* *Discrepancies between pre-training and fine-tuning* are revealed in Tab. 2 by setting

Table 3: **Ablation study of** $\sigma$. "$\sigma = 0$" means no position smoothing. "$\rightarrow$" denotes a linear decay schedule is performed on $\sigma$. We report the *averaged* top-1 accuracy of position predictions.

| $\sigma$ | ImageNet-1K | | COCO detection | | | COCO segmentation | | | ADE20k | |
| | fine-tune | position | $AP^b$ | $AP^b_{50}$ | $AP^b_{75}$ | $AP^m$ | $AP^m_{50}$ | $AP^m_{75}$ | mIoU | aAcc |
|---|---|---|---|---|---|---|---|---|---|---|
| 0 | 82.85 | 88.81 | 40.32 | 59.89 | 43.90 | 36.45 | 57.09 | 38.88 | 38.32 | 79.22 |
| 1 | 82.91 | 87.13 | 40.19 | 59.76 | 43.81 | 36.30 | 57.06 | 38.87 | 38.60 | 79.17 |
| 2 | 82.79 | 69.48 | 40.16 | 59.94 | 43.92 | 36.32 | 57.32 | 38.88 | 38.69 | 79.30 |
| $1 \rightarrow 0$ | **82.96** | 87.83 | **42.14** | 61.99 | 46.38 | **37.93** | 59.23 | 40.76 | **40.68** | 80.14 |
| $2 \rightarrow 0$ | 82.88 | 87.65 | 40.46 | 60.03 | 44.21 | 36.53 | 57.20 | 39.30 | 38.79 | 79.43 |

Table 4: **Ablation study of attentive reconstruction.** "$\tau = \infty$" indicates no attentive reconstruction. We report the *averaged* top-1 accuracy of position predictions.

| $\tau$ | ImageNet-1K | | COCO detection | | | COCO segmentation | | | ADE20k | |
| | fine-tune | position | $AP^b$ | $AP^b_{50}$ | $AP^b_{75}$ | $AP^m$ | $AP^m_{50}$ | $AP^m_{75}$ | mIoU | aAcc |
|---|---|---|---|---|---|---|---|---|---|---|
| $\infty$ | 82.84 | 88.66 | 40.56 | 60.48 | 44.15 | 36.78 | 57.64 | 39.46 | 38.38 | 79.36 |
| 0.1 | **82.96** | 87.83 | **42.14** | 61.99 | 46.38 | **37.93** | 59.23 | 40.76 | **40.68** | 80.14 |
| 0.5 | 82.86 | 87.78 | 40.73 | 60.33 | 44.77 | 36.71 | 57.72 | 39.38 | 38.80 | 79.62 |

$\gamma_{\text{pos}} = 1$, where we observe performance deterioration. ***(iii) Deciding which position to reconstruct precisely is crucial*** because by setting $\sigma = 0$ in Tab. 3 and $\tau = \infty$ in Tab. 4, ViTs have better position predictions task but perform worse on downstream tasks.

**Performance on the position reconstruction task.** In general, a *positive correlation* is observed between the accuracy of the downstream task and the averaged accuracy of predicted positions, indicating that the averaged accuracy of position predictions is a reliable indicator of how well the model is trained. However, there are some exceptions, *i.e.*, $\sigma = 0$ in Tab. 3 and $\tau = \infty$ in Tab. 4, where the model reconstructs more precise positions but performs worse on downstream benchmarks, indicating that *deciding which position to reconstruct precisely is crucial*.

Please refer to *Supplementary Material* for the detailed performance of the position reconstruction task, where we provide more evidence to support the three difficulties proposed in Sec. 1.

**Patch mask ratio** $\gamma$. We vary the patch mask ratio $\gamma$ from 0 to 0.75 in Tab. 1. When $\gamma = 0$, the entire image is visible during pre-training, resulting in nearly *perfect* position predictions and making the task too simple for self-supervised learning (see *Supplementary Material* for details). However, the *averaged* accuracy of position predictions is quite low since the accuracy deteriorates quickly as we enlarge $\gamma$ during evaluation. Adopting a larger $\gamma$ for pre-training leads to better performance, especially on detection and segmentation benchmarks. Therefore, we can conclude that *patch masking is essential in DropPos to prevent trivial solutions*. However, deterioration is observed when $\gamma = 0.9$. This is because the model may be under-fitted using this extremely difficult pretext task.

**Position mask ratio** $\gamma_{\text{pos}}$. We study the effectiveness of the position mask ratio $\gamma_{\text{pos}}$ in Tab. 2. DropPos appears to be more *robust* against different $\gamma_{\text{pos}}$ than $\gamma$. However, an appropriate $\gamma_{\text{pos}}$ is still necessary. A small $\gamma_{\text{pos}}$ leads to trivial solutions even if we have a large $\gamma$, while an extremely large $\gamma_{\text{pos}}$, *e.g.*, $\gamma_{\text{pos}} = 1$, results in *discrepencies between pre-training and fine-tuning*.

**Position smoothing.** By setting different values of $\sigma_0$ and $\sigma_T$, we study the effectiveness of position smoothing. By setting $\sigma = 0$, we remove the position smoothing strategy, and the model tends to be overwhelmed by this position reconstruction task, leading to *high accuracy in position prediction but poor performance on downstream tasks*. However, when $\sigma$ is too large, the smoothed positions become noisy, contributing to *poor* performance on *both* downstream tasks and position reconstruction. Additionally, equipped with a linear decay schedule of $\sigma$, DropPos is gradually guided, bringing improvements on *both* downstream tasks and position reconstruction.

**Attentive reconstruction.** Attentive reconstruction is studied in Tab. 4 by setting different values of temperature $\tau$, where $\tau = \infty$ means no attentive reconstruction. Without attentive reconstruction, DropPos is able to obtain better position predictions but performs worse on downstream tasks. This is because we do *not* have to reconstruct the *exact* positions when *different patches share similar visual appearances*. This phenomenon can be mitigated by setting an appropriate $\tau$. A large $\tau$ leads

Table 5: **Comparison with previous methods on ImageNet-1K classification.** All methods are evaluated by fine-tuning. The resolution of images is fixed to 224×224. † means our implementation. ‡ means the result is borrowed from [28].

| method | venue | eff. ep. | ViT-B | ViT-L |
|---|---|---|---|---|
| supervised | | - | 80.9[†] | 82.6[‡] |
| MAE [28] | [CVPR'22] | 200 | 82.2[†] | 83.3[‡] |
| DropPos | [Ours] | 200 | **83.0** | **83.7** |
| *Contrastive Learning* | | | | |
| DINO[‡] [6] | [ICCV'21] | 1600 | 82.8 | - |
| MoCo v3[‡] [13] | [ICCV'21] | 600 | 83.2 | 84.1 |
| *Masked Image Modeling* | | | | |
| BEiT[‡] [3] | [ICLR'22] | 800 | 83.2 | 85.2 |
| MAE[‡] [28] | [CVPR'22] | 1600 | 83.6 | **85.9** |
| SimMIM [64] | [CVPR'22] | 800 | 83.8 | - |
| SemMAE [34] | [NeurIPS'22] | 800 | 83.4 | - |
| LocalMIM [56] | [CVPR'23] | 1600 | 84.0 | - |
| HPM [54] | [CVPR'23] | 800 | **84.2** | 85.8 |
| *Masked Image Modeling + Contrastive Learning* | | | | |
| iBOT [72] | [ICLR'22] | 1600 | 84.0 | - |
| BootMAE [19] | [ECCV'22] | 800 | **84.2** | **85.9** |
| *Position Reconstructing* | | | | |
| DropPos | [Ours] | 800 | **84.2** | 85.8 |

Table 6: **Comparison with previous methods on downstream tasks.** All methods take the ViT-B/16 [21] as the backbone and utilize Mask R-CNN [30] on COCO [38] object detection and instance segmentation, and UperNet [63] on ADE20k [71] semantic segmentation, respectively. ‡ means the result is borrowed from [28]. † indicates our implementation, including pre-training and supervised fine-tuning, while ♯ represents we reproduce fine-tuning using the official pre-trained backbone. We perform 1× schedule of fine-tuning on COCO using ViTDet [36], and 80k iterations of fine-tuning on ADE20k using mmsegmentation [14].

| method | venue | eff. ep. | COCO detection | | | COCO segmentation | | | ADE20k | |
|---|---|---|---|---|---|---|---|---|---|---|
| | | | $AP^b$ | $AP^b_{50}$ | $AP^b_{75}$ | $AP^m$ | $AP^m_{50}$ | $AP^m_{75}$ | mIoU | aAcc |
| MAE[†] [28] | [CVPR'22] | 200 | 40.1 | 60.5 | 44.1 | 36.4 | 57.8 | 39.3 | 40.5 | 80.1 |
| DropPos | [Ours] | 200 | **42.1** | 62.0 | 46.4 | **37.9** | 59.2 | 40.8 | **40.7** | 80.1 |
| MoCo v3[♯] [13] | [ICCV'21] | 600 | 43.7 | 65.7 | 47.7 | 39.1 | 62.0 | 41.8 | 44.7 | 81.5 |
| MAE[♯] [28] | [CVPR'22] | 1600 | 47.3 | 68.2 | 52.5 | 42.4 | 65.3 | 45.6 | 47.0 | 82.7 |
| BootMAE[♯] [19] | [ECCV'22] | 800 | 47.3 | 67.9 | 52.1 | 42.3 | 65.0 | 45.8 | 47.3 | 83.0 |
| SemMAE[♯] [34] | [NeurIPS'22] | 800 | 45.6 | 66.2 | 55.2 | 40.9 | 63.3 | 44.4 | 44.9 | 82.0 |
| LocalMIM[♯] [56] | [CVPR'23] | 1600 | 47.4 | 67.7 | 52.2 | 42.2 | 64.8 | 45.5 | 47.1 | 83.1 |
| DropPos | [Ours] | 800 | **47.7** | 68.3 | 52.8 | **42.6** | 65.3 | 46.2 | **47.8** | 82.8 |

to noisy position ground truths, leading to *poor* performance on *both* downstream tasks and position reconstruction, which is the same as a large $\sigma$.

## 4.2 Comparisons with previous results

We compare our proposed DropPos with the supervised pre-training baseline and a wide range of self-supervised methods, including *(i)* contrastive learning methods [6, 13], *(ii)* masked image modeling methods [3, 28, 34, 54, 56, 64], and *(iii)* their combinations [19, 72]. Effective pre-training epoch is used for fair comparison following [54, 72] since it accounts for the *actual* trained images/views. The detailed definition can be found in *Supplementary Material*. All methods are pre-trained with the same resolution, *i.e.*, 224×224 on ImageNet-1K [48].

**ImageNet-1K classification.** We compare our DropPos with state-of-the-art alternatives on the ImageNet-1K [48] classification benchmark in Tab. 5. Notably, with only 200 epochs pre-training,

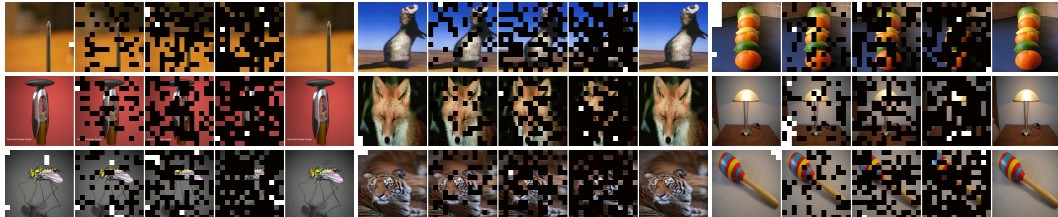

Figure 3: **Qualitative results of position reconstruction.** We evaluate position predictions with *different* $\gamma$ but fix $\gamma_{\text{pos}} = 0.95$. Black patches are *masked* during inference. The positions of those white patches are wrongly predicted, while the remaining patches are predicted correctly. For each tuple, we show results under **(a)** $\gamma = 0$, **(b)** $\gamma = 0.25$, **(c)** $\gamma = 0.5$, **(d)** $\gamma = 0.75$, and **(e)** the original image. DropPos manages to reconstruct *precise* positions.

Table 7: An in-depth analysis about **whether the improved position sensitivity benefits downstream tasks or not**. We freeze the backbone and train an extra linear patch classification head. 75% of position embeddings are randomly masked during linear probing. We can conclude that *image classification indeed needs better location awareness* and thus designing a position prediction pretext task is crucial and worth studying.

| method | venue | Position prediction | | ImageNet-1K |
| | | Pre-trained | Fine-tuned | Top-1 |
|---|---|---|---|---|
| MoCo v3 [13] | [ICCV'21] | 43.2 | 78.0 ↑ 34.8 | 83.2 |
| MAE [28] | [CVPR'22] | 63.1 | 82.5 ↑ 19.4 | 83.6 |
| DropPos | [ours] | **77.3** | **88.0** ↑ 10.7 | **84.2** |

DropPos achieves 83.0% and 83.7% using ViT-B/16 and ViT-L/16 as the backbone, surpassing the supervised baseline by +1.1%, and MAE [28] by +0.8% and +0.4% respectively. This empirical evidence demonstrates that *enhancing the location awareness of ViTs indeed brings better visual representations*. Furthermore, with 800 epochs of pre-training, DropPos manages to achieve competitive results compared with the state-of-the-art. Specifically, it achieves 84.2% and 85.8% using ViT-B/16 and ViT-L/16, respectively. Strikingly, DropPos reaches competitive results with BootMAE [19], which combines contrastive learning and masked image modeling.

**COCO object detection and segmentation.** We fine-tune Mask R-CNN [30] on COCO [38] with $1\times$ schedule, *i.e.*, 12 epochs, using the configuration of ViTDet [36]. We take ViT-B/16 [21] as the backbone for all entries in Tab. 6. We regard $\text{AP}^{\text{b}}$ and $\text{AP}^{\text{m}}$ as the main metric for object detection and instance segmentation, respectively. For further comparison, we additionally report $\text{AP}^{\text{b}}_{50}$ and $\text{AP}^{\text{b}}_{75}$ for object detection, and $\text{AP}^{\text{m}}_{50}$ and $\text{AP}^{\text{m}}_{75}$ for instance segmentation. With only 200 epochs of pre-training, DropPos achieves 42.1% $\text{AP}^{\text{b}}$ and 37.9% $\text{AP}^{\text{m}}$, outperforming MAE [28] by +2.0% and +1.5%, respectively. Note that these improvements appear to be more significant than those on the classification benchmark shown in Tab. 5, indicating that *DropPos indeed contributes to better spatial reasoning abilities of ViTs*. With 800 epochs of pre-training, DropPos achieves 47.7% $\text{AP}^{\text{b}}$ and 42.6% $\text{AP}^{\text{m}}$, surpassing MAE [28] by +0.4% and +0.2%, respectively.

**ADE20k semantic segmentation.** We fine-tune UperNet [63] on ADE20k [71] with 80k iterations. We take ViT-B/16 [21] as the backbone for all entries in Tab. 6 and search for the optimal learning rate for each entry. We regard mIoU as the main metric for semantic segmentation. We additionally report aAcc for further comparison. With only 200 epochs of pre-training, DropPos achieves 40.7% mIoU, outperforming MAE [28] by +0.2%. With 800 epochs of pre-training, DropPos achieves 47.8% mIoU, surpassing MAE [28] by +0.8%.

**Qualitative results.** We load DropPos with 200 epochs of pre-training and provide qualitative results on this position reconstruction task in Fig. 3. DropPos manages to reconstruct the exact positions of most patches *even in extreme circumstances*, *i.e.*, $\gamma = 0.75$ and $\gamma_{\text{pos}} = 0.95$. This suggests that DropPos, as a self-supervised pretext task, *indeed makes ViTs location-aware*.

### 4.3 Analysis

To explore whether the improved position sensitivity results in better feature representation and benefits to downstream tasks or not, we propose a metric to evaluate the position sensitivity.

Specifically, we freeze the backbone and train an extra linear position prediction head using the vanilla cross-entropy loss. Top-1 accuracies of position predictions *before and after* fine-tuning are reported in Tab. 7, and 75% of position embeddings are randomly masked during training. Higher values mean the model is better at modeling the position relationship. The top-1 accuracy on the ImageNet-1K [48] validation set after fine-tuning is also reported.

As shown in Tab. 7, the backbone performs better in position prediction after fine-tuning, indicating that *image classification indeed needs strong abilities in modeling spatial relationships*. It means that *better position sensitivity corresponds to better performances on downstream tasks*. This evidence suggests that our motivation, *i.e.*, enhancing the location awareness of ViTs, is reasonable, and the topic is worth studying. By designing a position prediction pretext task, the backbone pre-trained by DropPos has better position modeling abilities, performing better on a variety of downstream tasks.

## 5  Conclusion

In this paper, we present DropPos, a novel, simple, and effective self-supervised pretext task that enhances the location awareness of ViTs. By masking patch tokens first and dropping positional embeddings next, ViTs are requesting to classify the position of each visible patch among all candidates based on partial visual appearance and a few anchor patches. In this way, we manage to avoid *(i) discrepancies* between pre-training and fine-tuning, *(ii) trivial solutions*, and *(iii)* reconstructing precise positions when *unnecessary*, resulting in improved spatial reasoning and understanding abilities of ViTs. Experiments across various benchmarks demonstrate the efficacy of DropPos, where DropPos *consistently* achieves competitive results compared with previous methods.

**Discussion.** Due to limited computational resources, we do not use DropPos to pre-train larger ViTs, *e.g.*, ViT-H/14. Despite these limitations, a growing need to design a pretext task that effectively uncovers the representation learning capabilities of vision models is becoming evident. We hope our DropPos will inspire future work. Additionally, how to enhance location awareness of vision models for spatial reasoning tasks in a supervised manner is also valuable to study.

## Acknowledgements

This work was supported in part by the Major Project for New Generation of AI (No. 2018AAA0100400), the National Natural Science Foundation of China (No. 61836014, No. U21B2042, No. 62072457, No. 62006231), and the InnoHK program.

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

# Supplementary Material

In this supplementary material, we first provide mode implementation details for reproducibility in Sec. A. Next, we provide more experiments in Sec. B. Finally, in Sec. C, we evaluate the performance of the position reconstruction task using pre-trained models under different settings, and we provide more evidence to support the proposed three difficulties in Sec. 1.

## A  Implementation details

**ViT architecture.** We follow the standard vanilla ViT [21] architecture used in MAE [28] as the backbone, which is a stack of Transformer blocks [51]. Following MAE [28], we use the fixed 2D sine-cosine positional embeddings during pre-training. For the downstream classification task, we use features globally averaged from the encoder output for both end-to-end fine-tuning.

**Effective training epochs.** Following iBOT [72], we take the effective training epochs as the metric of the training schedule, due to extra computation costs brought by the multi-crop [5] augmentation, which is a widely used technique for contrastive methods. Specifically, the effective training epochs are defined as the actual pre-training epochs multiplied with a scaling factor $r$. For instance, DINO [6] is trained with 2 global 224×224 crops and 10 local 96×96 crops, and thus $r = 2 + (96/224)^2 \times 10 \approx 4$. More details and examples can be found in [72].

### A.1  ImageNet classification

For all experiments in this paper, we take ImageNet-1K [48], which contains 1.3M images for 1K categories, as the pre-trained dataset. By default, we take ViT-B/16 [21] as the backbone and it is pre-trained 200 epochs followed by 100 epochs of end-to-end fine-tuning. Implementation details can be found in the following table. Most of the configurations are borrowed from MAE [28]. The linear learning rate scaling rule is adopted: $lr = lr_{\text{base}} \times \text{batch\_size} / 256$. For supervised training from scratch, we simply follow the fine-tuning setting without another tuning. For ViT-B/16, pre-training and fine-tuning are conducted with 64 and 32 2080Ti GPUs, respectively. For ViT-L/16, pre-training and fine-tuning are conducted with 32 and 16 Tesla V100 GPUs, respectively.

| config | pre-training | fine-tuning |
|---|---|---|
| optimizer | AdamW | AdamW |
| base learning rate | 1.5e-4 | 1e-3 |
| weight decay | 0.05 | 0.05 |
| momentum | $\beta_1, \beta_2 = 0.9, 0.95$ | $\beta_1, \beta_2 = 0.9, 0.999$ |
| layer-wise lr decay | 1.0 | 0.8 |
| batch size | 4096 | 1024 |
| learning rate schedule | cosine decay | cosine decay |
| warmup epochs | 10 (ViT-B/16), 40 (ViT-L/16) | 5 |
| training epochs | 200 | 100 (ViT-B/16), 50 (ViT-L/16) |
| augmentation | RandomResizedCrop | RandAug (9, 0.5) [16] |
| label smoothing | - | 0.1 |
| mixup [68] | - | 0.8 |
| cutmix [65] | - | 1.0 |
| drop path [31] | - | 0.1 |

### A.2  COCO object detection and segmentation

We take Mask R-CNN [30] with FPN [37] as the object detector. Following [28] and [54], to obtain pyramid feature maps for matching the requirements of FPN [37], whose feature maps are all with a stride of 16, we equally divide the backbone into 4 subsets, each consisting of a last global-window block and several local-window blocks otherwise, and then apply convolutions to get the intermediate feature maps at different scales (stride 4, 8, 16, or 32).

We perform end-to-end fine-tuning on COCO [38] for 1× schedule with 1024×1024 resolution, where 88,750 iterations of training with a batch size of 16 are performed. We simply follow the configuration of ViTDet [36], where the learning rate is 3e-4 and decays at the 78,889-th and 85,463-th iteration by a factor of 10. Experiments are conducted on 8 Tesla V100 GPUs.

Table S1: **Ablation study of the initialization of the positional encoding.** *Fixed* sin-cos position embeddings achieve the best performance.

| initialization | learnable | ImageNet Top-1 Acc | ADE20k mIoU |
|---|---|---|---|
| sin-cos | ✗ | **82.96** | **40.68** |
| sin-cos | ✓ | 82.81 | 39.37 |
| random | ✓ | 82.48 | 38.72 |

Table S2: **The performance when DropPos is equipped with Swin Transformers [40].** 200 epochs of pre-training and 100 epochs of fine-tuning are performed based on the implementation of UM-MAE [35]. We can conclude that *DropPos works with Swin* [40].

| method | backbone | ImageNet-1K |
|---|---|---|
| UM-MAE [35] | Swin-Tiny [40] | 82.04 |
| DropPos | Swin-Tiny [40] | **82.73** |

## A.3   ADE20k semantic segmentation

We take UperNet [63] as the segmentation decoder following the code of [3, 14, 54]. Fine-tuning on ADE20k [71] for 80k iterations is performed. Specifically, each iteration consists of 16 images with 512×512 resolution. The AdamW optimizer is adopted with an initial learning rate of 7e-4 and a weight decay of 0.05 with ViT-B. We apply a polynomial learning rate schedule with the first warmup of 1500 iterations following common practice [3, 14, 54]. When fine-tuning using backbones pre-trained with different methods, we search for the optimal learning rate or simply follow their official implementation for a fair comparison. Specifically, the learning rate is 1e-4 for [13, 28, 34], 4e-4 for [19, 56], respectively. All experiments are conducted on 8 Tesla V100 GPUs.

## B   More Experiments

**The initialization of the positional encoding.** DropPos uses fixed 2D sin-cos position embeddings by default. We ablate the initialization of position embeddings in Tab. S1 and it demonstrates that fixed sin-cos position embeddings achieve the best performance.

**DropPos with Swin Transformers [40].**   To verify the scaling property and the generalization of DropPos, we provide experiments when DropPos is equipped with the Swin Transformer. We follow the implementation of UM-MAE [35] and pre-train a Swin-Tiny [40] from scratch using DropPos. All models are pre-trained with 200 epochs and fine-tuned with 100 epochs, following the configuration of UM-MAE [35]. From Tab. S2, we can conclude that DropPos still works on Swin Transformers [40], and thus enhancing the location awareness of vision transformers is still worth studying.

## C   Performance of position reconstruction

In this section, we evaluate the performance of the position reconstruction task using pre-trained models under different settings. Specifically, we vary $\gamma \in \{0, 0.25, 0.5, 0.75\}$ and $\gamma_{\text{pos}} \in \{0.25, 0.5, 0.75, 0.95\}$ when measuring the position prediction accuracy. We report performance under different evaluation settings as well as the *averaged* accuracy among 16 different cases. From Tabs. S3 to S5, we find evidence to support the three difficulties for designing an appropriate position-related pretext task introduced in Sec. 1: *(i)* discrepancies between pre-training and fine-tuning, *(ii)* failing to learn highly semantic representations by solving this simple position reconstruction task, and *(iii)* difficult to decide which patch positions to reconstruct precisely.

We study the effectiveness of different values of $\gamma$ during pre-training in Tab. S3. Interestingly, we find evidence for *failing to learn highly semantic representations by solving this simple position reconstruction task.* As illustrated by Tab. S3a, the pre-trained model performs *extremely well* when we set $\gamma = 0$ for evaluation but fails to keep this trend when we enlarge $\gamma$. This indicates that given

Table S3: Top-1 accuracy of position reconstruction using ViT-B/16 [21] pre-trained with **different mask ratio** $\gamma$. We underline the special parameter different from our default settings. Default settings are highlighted in color. "Avg. acc" is the *averaged* top-1 accuracy over 16 different cases. We evaluate the performance using the *same* pre-trained model under different $\gamma$ and $\gamma_{\text{pos}}$. Larger $\gamma$ and $\gamma_{\text{pos}}$ indicates a more challenging task.

(a) Pre-training with $\underline{\gamma = 0}$ (Avg. acc: 59.79).

| $\gamma$ \ $\gamma_{\text{pos}}$ | 0.25 | 0.50 | 0.75 | 0.95 | avg. |
|---|---|---|---|---|---|
| 0.00 | 99.37 | 99.26 | 99.15 | 98.34 | 99.03 |
| 0.25 | 87.97 | 87.45 | 86.20 | 70.79 | 83.10 |
| 0.50 | 55.37 | 55.11 | 49.84 | 22.96 | 45.82 |
| 0.75 | 12.99 | 14.85 | 12.85 | 4.21 | 11.23 |

(b) Pre-training with $\underline{\gamma = 0.25}$ (Avg. acc: 79.62).

| $\gamma$ \ $\gamma_{\text{pos}}$ | 0.25 | 0.50 | 0.75 | 0.95 | avg. |
|---|---|---|---|---|---|
| 0.00 | 99.24 | 99.26 | 99.18 | 98.92 | 99.15 |
| 0.25 | 98.67 | 98.81 | 98.63 | 97.21 | 98.33 |
| 0.50 | 93.96 | 93.62 | 90.01 | 62.11 | 84.93 |
| 0.75 | 50.02 | 47.79 | 35.03 | 11.46 | 36.08 |

(c) Pre-training with $\gamma = 0.5$ (Avg. acc: 87.27).

| $\gamma$ \ $\gamma_{\text{pos}}$ | 0.25 | 0.50 | 0.75 | 0.95 | avg. |
|---|---|---|---|---|---|
| 0.00 | 99.33 | 99.23 | 99.13 | 98.85 | 99.14 |
| 0.25 | 99.05 | 98.95 | 98.78 | 98.20 | 98.75 |
| 0.50 | 94.60 | 95.94 | 94.28 | 83.31 | 92.03 |
| 0.75 | 78.77 | 72.75 | 59.54 | 25.59 | 59.16 |

(d) Pre-training with $\gamma = 0.75$ (Avg. acc: 87.83).

| $\gamma$ \ $\gamma_{\text{pos}}$ | 0.25 | 0.50 | 0.75 | 0.95 | avg. |
|---|---|---|---|---|---|
| 0.00 | 97.19 | 98.69 | 98.20 | 92.30 | 96.60 |
| 0.25 | 96.78 | 98.26 | 97.66 | 91.05 | 95.93 |
| 0.50 | 97.26 | 96.82 | 95.66 | 89.12 | 94.72 |
| 0.75 | 79.94 | 78.10 | 68.73 | 40.24 | 66.75 |

Table S4: Top-1 accuracy of position reconstruction using ViT-B/16 [21] pre-trained with **different positional mask ratio** $\gamma_{\text{pos}}$. We underline the special parameter different from our default settings.

(a) Pre-training with $\gamma_{\text{pos}} = 0.25$ (Avg. acc: 65.19).

| $\gamma$ \ $\gamma_{\text{pos}}$ | 0.25 | 0.50 | 0.75 | 0.95 |
|---|---|---|---|---|
| 0.00 | 96.52 | 89.38 | 59.29 | 14.58 |
| 0.25 | 96.62 | 91.27 | 66.00 | 19.17 |
| 0.50 | 95.58 | 91.10 | 72.07 | 21.69 |
| 0.75 | 80.56 | 73.72 | 57.89 | 17.66 |
| avg. | 92.32 | 86.37 | 63.81 | 18.28 |

(b) Pre-training with $\gamma_{\text{pos}} = 0.5$ (Avg. acc: 86.70).

| $\gamma$ \ $\gamma_{\text{pos}}$ | 0.25 | 0.50 | 0.75 | 0.95 |
|---|---|---|---|---|
| 0.00 | 98.98 | 98.81 | 98.24 | 92.80 |
| 0.25 | 98.59 | 98.34 | 97.51 | 89.73 |
| 0.50 | 96.63 | 95.85 | 93.40 | 74.38 |
| 0.75 | 81.94 | 76.70 | 64.87 | 30.44 |
| avg. | 94.04 | 92.43 | 88.51 | 71.84 |

(c) Pre-training with $\gamma_{\text{pos}} = 0.75$ (Avg. acc: 87.83).

| $\gamma$ \ $\gamma_{\text{pos}}$ | 0.25 | 0.50 | 0.75 | 0.95 |
|---|---|---|---|---|
| 0.00 | 97.19 | 98.69 | 98.20 | 92.30 |
| 0.25 | 96.78 | 98.26 | 97.66 | 91.05 |
| 0.50 | 97.26 | 96.82 | 95.66 | 89.12 |
| 0.75 | 79.94 | 78.10 | 68.73 | 40.24 |
| avg. | 92.79 | 92.97 | 90.06 | 78.18 |

(d) Pre-training with $\underline{\gamma_{\text{pos}} = 1}$ (Avg. acc: 19.44).

| $\gamma$ \ $\gamma_{\text{pos}}$ | 0.25 | 0.50 | 0.75 | 0.95 |
|---|---|---|---|---|
| 0.00 | 15.57 | 23.24 | 20.41 | 23.59 |
| 0.25 | 12.51 | 21.10 | 24.30 | 29.41 |
| 0.50 | 7.76 | 14.18 | 25.56 | 45.01 |
| 0.75 | 3.34 | 6.00 | 12.53 | 26.45 |
| avg. | 9.80 | 16.13 | 20.70 | 31.12 |

the strength of ViTs in modeling long-range dependencies, they have easily solved this task in a superficial way, and thus pre-training with $\gamma = 0$ becomes trivial for ViTs. To this end, an appropriate $\gamma$ is necessary to increase the difficulty of the pretext task and avoid trivial solutions.

We study the effectiveness of different values of $\gamma_{\text{pos}}$ during pre-training in Tab. S4, and we find evidence for *discrepancies between pre-training and fine-tuning.* As shown by Tab. S4d, the model fails to reconstruct accurate positions given some visible anchors. This is because the model has *never* been exposed to any positional embeddings (PEs) during pre-training. Therefore, providing some anchors is necessary to address discrepancies. Also, it may help the model focus on modeling *relative* relationships instead of simply reconstructing absolute positions.

Table S5: Top-1 accuracy of position reconstruction using ViT-B/16 [21] pre-trained with **different (i)** $\sigma$ **and (ii)** $\tau$. We underline the special parameter different from our default settings.

(a) Pre-training with $\underline{\sigma = 0}$ (Avg. acc: 88.81).

| $\gamma$ \ $\gamma_{\text{pos}}$ | 0.25 | 0.50 | 0.75 | 0.95 |
|---|---|---|---|---|
| 0.00 | 98.48 | 98.74 | 98.29 | 94.37 |
| 0.25 | 98.06 | 98.31 | 97.76 | 93.48 |
| 0.50 | 96.06 | 96.08 | 94.48 | 85.49 |
| 0.75 | 82.19 | 78.64 | 69.39 | 41.23 |

(b) Pre-training with $\underline{\sigma = 1}$ (Avg. acc: 87.13).

| $\gamma$ \ $\gamma_{\text{pos}}$ | 0.25 | 0.50 | 0.75 | 0.95 |
|---|---|---|---|---|
| 0.00 | 97.03 | 98.50 | 97.93 | 91.40 |
| 0.25 | 96.63 | 98.04 | 97.33 | 89.69 |
| 0.50 | 94.30 | 95.58 | 93.68 | 81.47 |
| 0.75 | 79.14 | 77.06 | 67.43 | 38.89 |

(c) Pre-training with $\underline{\sigma = 2}$ (Avg. acc: 69.48).

| $\gamma$ \ $\gamma_{\text{pos}}$ | 0.25 | 0.50 | 0.75 | 0.95 |
|---|---|---|---|---|
| 0.00 | 78.45 | 78.81 | 78.19 | 72.33 |
| 0.25 | 78.03 | 78.41 | 77.68 | 70.49 |
| 0.50 | 76.14 | 76.27 | 74.53 | 63.52 |
| 0.75 | 63.75 | 61.21 | 53.30 | 30.47 |

(d) Pre-training with $\sigma = 1 \rightarrow 0$ (Avg. acc: 87.83).

| $\gamma$ \ $\gamma_{\text{pos}}$ | 0.25 | 0.50 | 0.75 | 0.95 |
|---|---|---|---|---|
| 0.00 | 97.19 | 98.69 | 98.20 | 92.30 |
| 0.25 | 96.78 | 98.26 | 97.66 | 91.05 |
| 0.50 | 97.26 | 96.82 | 95.66 | 89.12 |
| 0.75 | 79.94 | 78.10 | 68.73 | 40.24 |

(e) Pre-training with $\underline{\sigma = 2 \rightarrow 0}$ (Avg. acc: 87.65).

| $\gamma$ \ $\gamma_{\text{pos}}$ | 0.25 | 0.50 | 0.75 | 0.95 |
|---|---|---|---|---|
| 0.00 | 97.63 | 98.61 | 97.93 | 91.30 |
| 0.25 | 97.22 | 98.19 | 97.46 | 89.96 |
| 0.50 | 94.99 | 95.85 | 94.06 | 82.50 |
| 0.75 | 80.33 | 77.92 | 68.42 | 40.11 |

(f) Pre-training with $\underline{\tau = \infty}$ (Avg. acc: 88.66).

| $\gamma$ \ $\gamma_{\text{pos}}$ | 0.25 | 0.50 | 0.75 | 0.95 |
|---|---|---|---|---|
| 0.00 | 98.66 | 98.31 | 97.72 | 91.22 |
| 0.25 | 98.54 | 98.17 | 97.46 | 91.00 |
| 0.50 | 96.85 | 96.20 | 94.52 | 84.64 |
| 0.75 | 83.57 | 79.30 | 70.04 | 42.29 |

(g) Pre-training with $\tau = 0.1$ (Avg. acc: 87.83).

| $\gamma$ \ $\gamma_{\text{pos}}$ | 0.25 | 0.50 | 0.75 | 0.95 |
|---|---|---|---|---|
| 0.00 | 97.19 | 98.69 | 98.20 | 92.30 |
| 0.25 | 96.78 | 98.26 | 97.66 | 91.05 |
| 0.50 | 97.26 | 96.82 | 95.66 | 89.12 |
| 0.75 | 79.94 | 78.10 | 68.73 | 40.24 |

(h) Pre-training with $\underline{\tau = 0.5}$ (Avg. acc: 87.78).

| $\gamma$ \ $\gamma_{\text{pos}}$ | 0.25 | 0.50 | 0.75 | 0.95 |
|---|---|---|---|---|
| 0.00 | 97.68 | 97.94 | 97.82 | 91.78 |
| 0.25 | 97.83 | 97.91 | 97.53 | 91.01 |
| 0.50 | 96.44 | 96.06 | 94.52 | 84.49 |
| 0.75 | 80.16 | 77.16 | 66.01 | 40.18 |

We study the effectiveness of different values of $\gamma_{\text{pos}}$ during pre-training in Tab. S4, and we find evidence for *hard to decide which patch positions to reconstruct precisely.* As shown by Tabs. S5a and S5f, the model achieves higher position prediction accuracy but performs worse on downstream tasks (please refer to Tabs. 3 and 4 for downstream performances). Therefore, to prevent being overwhelmed by this particular position reconstruction task, techniques for relaxing the patch-wise classification problem become necessary, *i.e.*, position smoothing, and attentive reconstruction.

