# OpenReview forum: "DropPos: Pre-Training Vision Transformers by Reconstructing Dropped Positions"
_NeurIPS.cc/2023/Conference — NeurIPS 2023 poster_

### Official Review · Reviewer_S2Rp · 2023-07-04

**Soundness:** 3 good
**Presentation:** 3 good
**Contribution:** 3 good
**Rating:** 6
**Confidence:** 4

**Summary:**

The paper introduces a novel self-supervised pretext task for Vision Transformers (ViTs), called DropPos. It aims to enhance the spatial reasoning or location awareness of ViTs, based on the observation that ViTs are often insensitive to the order of input tokens. DropPos works by dropping a large random subset of positional embeddings, then using the model to predict the actual position of each patch based solely on its visual appearance.

The paper identifies three major difficulties: (a) discrepancies between pre-training and fine-tuning (b) trivial solutions that fail to learn highly semantic representations by solving this simple task (c) patches with similar visual content. To prevent trivial solutions and increase task difficulty, this paper keeps only a subset of patches visible during the task. Given the potential similarity in visual appearances between different patches, the authors propose position smoothing and attentive reconstruction strategies. This relaxation allows for non-exact position reconstruction when exact positions are not critical. Quantitative evaluations demonstrate the effectiveness of DropPos, outperforming supervised pre-training and yielding competitive results against state-of-the-art self-supervised alternatives on various benchmarks.

**Strengths:**

1. The main claim is attractive. It would be quite interesting (and a bit counter-intuition) to see the vision transformers can learn a very good representation by such a simple patch predicting task, which is very coarse-grained. Particularly, the results of this paper look competitive to the mainstream pre-training tasks.

2. This work provides extensive ablation studies to verify their design and to reveal some insights.

3. It proposes Position smoothing and Attentive reconstruction to solve the problems like patches may share similar visual appearance.


**Weaknesses:**


1. The paper requires additional experiments and deeper analysis to substantiate some assertions.

For example, this paper starts from "Vision Transformers (ViTs) are quite insensitive to the order of input tokens, the need for an appropriate self-supervised pretext task that enhances the location awareness of ViTs is becoming evident.".

Although this claim fits my intuition, it is weird that after citing some papers [13, 39, 60], the paper does not talk about this claim any more in the following sections. What would happen if messing up the order of input tokens? Or even further, should this really be viewed as a drawback? Insensitiveness to the order may also be a good property. The authors touch on this in the related work, but this claim needs more attention since it's the main drive of the paper.

Also, it's unclear if DropPos helps this issue. More specifically, if a model is pre-trained by DropPos and then finetuned in ImageNet, would it be more sensitive to the order of input tokens? Looking into this would help us understand if the improved performance comes from the model being more "sensitive to the order of input tokens".

2. A recent work: Jigsaw-ViT: Learning jigsaw puzzles in vision transformer.

Another work Jigsaw-ViT proposes to include solving Jigsaw in the training of ViT, which is close to the task of this work. It is beneficial to include this for comparison.

[1] Chen, Yingyi, Xi Shen, Yahui Liu, Qinghua Tao, and Johan AK Suykens. "Jigsaw-ViT: Learning jigsaw puzzles in vision transformer." Pattern Recognition Letters 166 (2023): 53-60.


**Questions:**

I am not really sure so I put this in Questions instead of Weaknesses. After checking the main paper and supplementary, it seems the authors did not mention if the pre-training is conducted with DropPos only, or together with other pre-training tasks like in MAE. Could the authors answer whether DropPos is the only training objective?

If the answer is yes, the reviewer would hope the authors can discuss a bit about why such a very coarse-grained task can be more powerful than the counterparts. Does it mean the dense visual cues are not important in pre-training?


Overall I am feeling this paper is a good trial, although it may need more in-depth analysis and appropriate discussion about relevant works.

---

> ### Author Rebuttal · Authors · 2023-08-10
>
> We thank reviewer S2Rp for the valuable time and constructive feedback. Point-to-point responses
> are provided below.
>
> **Q1: Additional experiments and deeper analysis are required to verify the motivation.**
>
> **A1:** We observed that the improved position sensitivity results in better feature representation and
> benefits to downstream tasks. To verify this argument, we first propose a metric to evaluate the
> model’s position sensitivity. Specifically, we freeze the backbone and train an extra linear position
> prediction head using the vanilla cross-entropy loss. Top-1 accuracies of position predictions before
> and after fine-tuning are reported, and 75% of position embeddings are randomly masked during
> training. Higher values mean the model is better at modeling the position relationship. The top-1
> accuracy on the ImageNet validation set after fine-tuning is also reported.
>
> Please refer to Table 2 in the "global" response for detailed results.
>
> As shown in the table, the backbone performs better in position prediction *after* fine-tuning, indicating
> that image classification indeed needs strong abilities in modeling spatial relationships. It means that
> better position sensitivity corresponds to better performances on downstream tasks. This evidence
> suggests that our motivation, i.e., enhancing the location awareness of ViTs, is reasonable, and the
> topic is worth studying. By designing a position prediction pretext task, the backbone pre-trained by
> DropPos has better position modeling abilities, performing better on a variety of downstream tasks.
>
> **Q2: Lack of comparison with [a].**
>
> **A2:** We apologize for the missed comparison. JigsawViT [a] explores solving jigsaw puzzles as an
> auxiliary objective in ViT for supervised image classification. The authors have demonstrated that the
> proposed extra objective brings consistent and significant improvements in supervised tasks. However,
> DropPos aims to design a brand new self-supervised pretext task that enhances the location awareness
> of ViTs, which is totally different. Therefore, it is relatively hard to compare the performances of
> these two works under the same benchmark.
>
> **Q3: About the objective.**
>
> **A3:** Sorry for the ambiguity. DropPos uses only the cross-entropy loss mentioned in the manuscript,
> and the MSE loss used in MAE is not adopted. We will clarify this in our revised version.
>
> **Q4: The reason why such a very coarse-grained task can be more powerful than its counterparts.**
>
> **A4:** This is an interesting question. It may be because images are natural signals with heavy spatial
> redundancy [27]. Towards this issue, some interesting works explore the best target representations
> for masked image modeling, e.g., [36] and [52]. They empirically found that raw RGB pixels may
> not be the best choice. Using coarse-grained targets such as HoG [52] features even performs better.
> On the other hand, instance discrimination or contrastive learning is also a very coarse-grained task.
> It ignores the possibility of different samples belonging to the same category.
>
> Therefore, what matters in pre-training seems to be highly semantic clues, rather than dense cues. An
> appropriate pretext task is still worth exploring.
>
> **References**
>
> [a] Yingyi Chen et al. Jigsaw-ViT: Learning jigsaw puzzles in vision transformer. Pattern Recognition
> Letters 166 (2023): 53-60.

---

> > ### Comment · Reviewer_S2Rp · 2023-08-16
> >
> > Thank you for the clarification. After reading the comments and the rebuttal, I tend to keep the score.

---

### Official Review · Reviewer_6DGy · 2023-07-04

**Soundness:** 3 good
**Presentation:** 3 good
**Contribution:** 2 fair
**Rating:** 6
**Confidence:** 5

**Summary:**

This paper introduces a novel approach to self-supervised representation learning for vision transformers, focusing on enhancing their positional awareness. The authors proposed a new pretext task called DropPos, which involves reconstructing the positions of dropped tokens in partial observations. By leveraging DropPos, which is a spiritual adaptation of MAE, the authors consistently achieve superior results compared to the state-of-the-art (SOTA) techniques in standard downstream tasks, including image classification, object detection, and semantic segmentation.

**Strengths:**

The paper effectively addresses the problem of incorporating inductive bias into vision transformers, and it takes an interesting approach by employing self-supervised learning (SSL) techniques. Enhancing the positional awareness of vision transformers is a significant aspect, as it can greatly improve their performance in downstream tasks that rely on location awareness and positional information. The proposed pretext task, DropPos, is simple yet effective, which aligns with the requirements of SSL. Additionally, such pretext tasks are efficient as they involve masking a substantial portion of the input data. The paper highlights that using tasks like DropPos eliminates the need for careful selection of target representation and mask strategy, as typically performed in mask image modeling. Furthermore, the paper is well-written and easy to follow.

**Weaknesses:**

One potential weakness of the paper lies in its experimental evaluation. While the proposed pretext task is commended for its simplicity and effectiveness, it would have been valuable to compare the performance of a DropPos pretrained Vision Transformer (ViT) with a hierarchical transformer such as Swin, pretrained using mask image modeling [A, B]. This comparison would have provided insights into whether a carefully designed ViT architecture already addresses the need for positional awareness, rendering the pretext task redundant.

Furthermore, the paper lacks an in-depth analysis of what the ViT is actually learning and how it achieves positional awareness. Quantifying the extent of positional-aware representation learning is crucial. Analysis such as intertoken distance within a layer, sparsity of attention weights, and linear probe results are missing, which would have shed light on the underlying reasons for the success of the proposed pretext task.

A significant concern arises in the evaluation section where the authors reproduce most of the state-of-the-art (SOTA) results reported in other papers. However, the reproduced numbers are generally lower than the original reported results, without any mention of the differences. It would be important for the authors to investigate the causes of these discrepancies, considering factors such as differences in the experimental setup (e.g., number of GPUs, changes in batch size and learning schedules) or potential missing engineering tricks. Particularly concerning is the lack of significant improvement (wrt MAE) in object detection and segmentation results, which are crucial for spatial modeling.

Lastly, the authors missed referencing relevant literature on contrastive learning approaches designed for spatial modeling, such as [C]. In fact, [C] outperforms DropPos when trained for 200 epochs on the ADE20k dataset.

[A] Zhenda Xie, Zheng Zhang, Yue Cao, Yutong Lin, Jianmin Bao, Zhuliang Yao, Qi Dai and Han Hu. SimMIM: A Simple Framework for Masked Image Modeling
[B] Xiang Li, Wenhai Wang, Lingfeng Yang, Jian Yang. Uniform Masking: Enabling MAE Pre-training for Pyramid-based Vision Transformers with Locality.
[C] Patch-level Representation Learning for Self-supervised Vision Transformers, CVPR 2022

**Questions:**

1. Why do we need to compute Affinity as indicated in equation (9). Why can't we use the self-attention matrix (softmax(KQ'))?
2. In Table 3, it will be also interesting to check the results of 2->1.
3. What happens when ViT is pretrained with DropPos for longer epochs like 1600 epochs?
4. In fig 3, how do we know that which predictions are correct?
For rebuttal, please also refer to the points mentioned in the weakness section.

**Limitations:**

The paper acknowledges a limitation in its conclusion, which the reviewer agrees with. Although the experiments conducted with ViT-B are deemed sufficient to demonstrate the potential of the proposed method, it is crucial to provide a comprehensive explanation and understanding of the model, regardless of its size. This ensures a robust analysis and comprehension of the proposed approach beyond the specific ViT-B architecture.

---

> ### Author Rebuttal · Authors · 2023-08-10
>
> We thank reviewer 6DGy for the valuable time and constructive feedback. Point-to-point responses
> are provided below.
>
> **Q1: DropPos with Swin.**
>
> **A1:** We provide experiments when DropPos is equipped with Swin. We follow the implementation of
> UM-MAE [a] and pre-train a Swin-Tiny from scratch. Please refer to Table 1 in the "global" response
> for detailed results. From the table, we can tell that *even a carefully designed ViT architecture has
> not addressed the need for positional awareness yet.*
>
> **Q2: An in-depth analysis and the linear probing performance.**
>
> **A2:** We answer this question in the following three perspectives.
>
> (1) Following your suggestion, we illustrate the sparsity of attention weights and the inter-token
> distance within a layer in Figure 1 and Figure 2 of the "global" response, respectively.
> - Figure 1 demonstrates that compared with MAE, *features of DropPos tend to have more sparse attention maps, especially at shallow layers.*
> - In Figure 2, we can conclude that compared with MAE, the shallow features of DropPos (depth 2 to 8) have lower distances, indicating smaller receptive fields. It means that *local patch relationships are more informative to help discriminate positions and enhance location awareness.*
>
> (2) To provide an in-depth analysis, we propose a metric to evaluate the model’s position sensitivity
> and explore the relationship between position sensitivity and performance on downstream tasks.
> Specifically, we freeze the backbone and train an extra linear position prediction head using the
> vanilla cross-entropy loss. Top-1 accuracies of position predictions are reported, and 75% of position
> embeddings are randomly masked during training. Higher values mean the model is better at modeling
> the position relationship. Please refer to Table 2 in the "global" response for detailed results. As
> shown in the table, the backbone performs better in position prediction after fine-tuning, indicating
> that *image classification indeed needs strong abilities in modeling spatial relationships*. It means that
> *better position sensitivity corresponds to better performances on downstream tasks.* By designing a
> position prediction pretext task, the backbone pre-trained by DropPos has better position modeling
> abilities, performing better on a variety of downstream tasks.
>
> (3) The linear probing accuracy of DropPos is 43.45% (ViT-B).
>
> **Q3: About the reproduced numbers, the insignificant improvements over MAE on detection
> and segmentation.**
>
> **A3:** The difference is due to the training iteration. For detection and segmentation tasks, we first
> download the pre-trained backbone and then perform end-to-end fine-tuning using the configuration
> of ViTDet [33] and mmsegmentation [14] for COCO and ADE20k experiments, respectively. *For
> efficient training, we perform 12 epochs of fine-tuning instead of 100 epochs on COCO, and 80k
> iterations instead of 160k iterations on ADE20k*, following [48] and [18]. Therefore, although the
> reproduced numbers are lower than their original numbers, we have conducted a fair comparison. We
> will clarify this and try to conduct experiments with longer schedules in our revised version.
>
> We would like to point out that the improvement over MAE on the ADE20k semantic segmentation
> benchmark is significant (+0.8 mIoU). As for COCO experiments, the improvements seem to be
> a little bit incremental may be because ViTDet [33] was originally tuned based on MAE, and we
> did not tune any parameters. However, consistent improvements over MAE are observed on COCO
> benchmarks, verifying the effectiveness of DropPos.
>
> **Q4: Lack of comparison with [b].**
>
> **A4:** We will add a brief discussion on contrastive learning approaches designed for spatial modeling
> in our revised version. As for the performances, [b] used the multi-crop augmentation technique.
> Therefore, the effective pre-training epoch should be $200 \cdot \frac{2 \cdot 224^2 + 8 \cdot 96^2}{224^2} \approx 700$ instead of 200.
> Simply comparing DropPos pre-trained with only 200 epochs seems to be unfair.
>
> **Q5: Affinity instead of self-attention.**
>
> **A5:** The self-attention map is a reasonable alternative. However, we empirically found that using
> affinity brings slightly better performances than using the self-attention map.
>
> **Q6: More experiments in Table 3.**
>
> **A6:** Appreciate! We conduct the suggested experiment. When $\sigma = 1 \to 0$, DropPos achieves 82.68 top-1 on ImageNet and 39.97 mIoU on ADE20k. This evidence indicates that reconstructing precise positions at the end of training is beneficial.
>
> **Q7: DropPos with 1600 epochs.**
>
> **A7:** We fail to pre-train a backbone with such a long schedule during the short rebuttal period due
> to limited computational resources. However, we hypothesize that DropPos is expected to perform
> better with a longer schedule, as the top-1 accuracy of position predictions is around 96% and still
> has room to improve. We will try to add this experiment in the future.
>
> **Q8: About Figure 3.**
>
> **A8:** We apologize for the ambiguity. Figure 3 shows the qualitative results of position reconstruction
> under different mask ratios γ. Black patches are masked during inference. The positions of those
> white patches are wrongly predicted, while the remaining patches are predicted correctly. From the
> figure, we can conclude that DropPos manages to reconstruct precise positions even under extremely
> difficult situations (e.g., γ = 0.75).
>
> **References**
>
> [a] Xiang Li et al. Uniform Masking: Enabling MAE Pre-training for Pyramid-based Vision
> Transformers with Locality. arXiv:2205.10063, 2022.
>
> [b] Sukmin Yun et al. Patch-level Representation Learning for Self-supervised Vision Transformers.
> In CVPR 2022.

---

> > ### Comment · Reviewer_6DGy · 2023-08-16
> > **Reproduced numbers**
> >
> > The reviewer acknowledges and appreciates the authors' efforts in addressing all raised concerns. It's acceptable to not include the model results from 1600 epochs. The incorporation of the Swin results, coupled with the in-depth analysis, will undoubtedly enhance the quality of the paper.
> >
> > The reviewer accepts the model configuration for the COCO and ADE experiments as stated: "For efficient training, we perform 12 epochs of fine-tuning instead of 100 epochs on COCO, and 80k iterations instead of 160k iterations on ADE20k, following [48] and [18]."
> >
> > However, there seems to be a discrepancy between the reported figures in this paper and those presented in [48] and [18]. This discrepancy has not been explained. For instance, the BootMAE results appear to have been under-reported. Additionally, it should be noted that the MAE results (trained for 1600 epochs) in the BootMAE paper outperform those of Droppos.

---

> > > ### Author Response · Authors · 2023-08-16
> > >
> > > We appreciate your response! We hope our clarifications below can address your concerns better.
> > >
> > > First of all, we would like to clarify that HPM [48] and BootMAE [18] performed 160k iterations of fine-tuning for ADE20k semantic segmentation results, while we only adopt 80k iterations of fine-tuning.
> > >
> > > Therefore, the discrepancy between the reproduced number and the reported number of BootMAE mainly lies in the results of COCO experiments.
> > > Specifically, the reported numbers are 48.5 box AP and 43.4 mask AP, while the reproduced numbers are 47.3 box AP and 42.3 mask AP.
> > > The reason should be the different code base. BootMAE was originally built using mmdetection [A], while the implementation of our DropPos is based on detectron2 [B]. The main difference between these two implementations is the input resolution. The input images of BootMAE are resized so that the shorter side is 800 pixels, while the longer side does not exceed 1333 pixels (as mentioned in the last paragraph of page 22), while DropPos takes 1024x1024 images as inputs (see [here](https://github.com/facebookresearch/detectron2/blob/main/projects/ViTDet/configs/common/coco_loader_lsj.py) for details).
> > >
> > > We recognize that there seems to be a significant difference. However, as the detection code of BootMAE has not been made publicly available, and the default configuration of ViTDet in mmdetection (it takes 1024x1024 as the input resolution, see [here](https://github.com/open-mmlab/mmdetection/blob/main/projects/ViTDet/configs/lsj-100e_coco-instance.py) for details) is different from that of BootMAE, it is relatively hard for us to have a detailed check.
> > >
> > > **References**
> > >
> > > [A] Kai Chen et al. MMDetection: Open mmlab detection toolbox and benchmark. arXiv preprint arXiv:1906.07155, 2019.
> > >
> > > [B] Yuxin Wu et al. Detectron2. https://github.com/facebookresearch/detectron2, 2019.
> > >
> > > Thanks again for your time and consideration. Please let us know if you have any questions. We are always looking forward to an open dialog.

---

> > > > ### Comment · Reviewer_6DGy · 2023-08-16
> > > > **Thanks for the clarification.**
> > > >
> > > > Ok, now this is clear to me. Thanks for the clarification.

---

### Official Review · Reviewer_HkQR · 2023-07-06

**Soundness:** 3 good
**Presentation:** 4 excellent
**Contribution:** 3 good
**Rating:** 7
**Confidence:** 4

**Summary:**

This paper presents a simple yet effective approach for generative self-supervised representation learning on images, namely DropPos.  The proposed approach drops a large random subset of positional embeddings for visible tokens and classifies the actual position for these tokens via visual appearance. Experimentally, DropPos outperforms state-of-the-art self-supervised approaches on a wide range of downstream benchmarks including image classification, detection and segmentation.

**Strengths:**

- The paper is well written and organized.
- The idea in this paper is simple yet effective, which brings something new in generative self-supervised representation learning.
- The authors provide clear implementation details (e.g., Pseudo-Code), which makes it easier to be reproduced.
- Detailed ablation study is conducted to verify the effectiveness of the proposed approach.
- Achieving SOTA performance on various downstream tasks.

**Weaknesses:**

- As illustrated in (4), the cross-entropy loss is applied for dropped position supervision. Except for this loss, is additional loss used? e.g., the MSE reconstruction loss used in MAE.
- In MAE, more layers are used in the decoder (i.e., 8). Here the decoder only consists of 2 layers. Is it because the dropped position classification task is easier than the patch reconstruction task?
- What’s the overall training time for DropPos? Is it comparable or more efficient than existing self-supervised approaches?
- For completeness, please add more dropout-guided work for discussion, e.g., DropMAE: Masked Autoencoders with Spatial-Attention Dropout for Tracking Tasks (CVPR23), which similarly employs dropout mechanism, but for spatial-attention dropout in videos.

**Questions:**

No. See Weaknesses.

**Limitations:**

No.

---

> ### Author Rebuttal · Authors · 2023-08-09
>
> We thank reviewer HkQR for the valuable time and constructive feedback. Point-to-point responses
> are provided below.
>
> **Q1: About the objective.**
>
> **A1:** DropPos uses only the cross-entropy loss mentioned in the manuscript, and the MSE loss used in
> MAE is not adopted. We will clarify this in our revised version.
>
> **Q2: About the decoder.**
>
> **A2:** In fact, when using decoders with different depths, the fine-tuning accuracy of MAE is almost
> the same (please refer to Table 1a in MAE [27]). We adopt a shallower decoder simply because it is
> more efficient. Moreover, we provide an ablation over the decoder depth in the following table. We
> take ViT-B as the backbone and all models are pre-trained with 200 epochs. From the table, we can
> tell that DropPos appears to be robust against different decoder depths.
>
> | # blocks | ImageNet | ADE20k |
> |---|---|---|
> | 2 | 82.96 | 40.68 |
> | 8 | 82.88 | 40.05 |
>
> **Q3: About the overall training time.**
>
> **A3:** The training procedure of each iteration is as efficient as MAE, and the overall training time is
> half that of MAE since DropPos is pre-trained with only 800 epochs.
>
> **Q4: Lack of discussion with dropout-guided works.**
>
> **A4:** Appreciate! We provide a brief discussion with representative dropout-guided studies in the
> following. [a] adaptively performs spatial-attention dropout in the frame reconstruction to facilitate
> temporal correspondence learning in videos, leading to a stronger temporal matching learner in visual
> object tracking and segmentation. [b] adopts feature-level dropout to the common weak-to-strong
> pipeline in semi-supervised semantic segmentation, bringing a broader perturbation space and thus
> resulting in better performances.
>
> To the best of our knowledge, DropPos is the first work that proposes a brand new self-supervised
> pretext task using dropout on position embeddings.
>
> **References**
>
> [a] Qiangqiang Wu et al. DropMAE: Masked Autoencoders with Spatial-Attention Dropout for
> Tracking Tasks. In CVPR 2023.
>
> [b] Lihe Yang et al. Revisiting Weak-to-Strong Consistency in Semi-Supervised Semantic
> Segmentation. In CVPR 2023.

---

> > ### Comment · Reviewer_HkQR · 2023-08-19
> >
> > Thanks for addressing my concerns in a sufficient way. I would like to keep my former decision.

---

### Official Review · Reviewer_e3Vr · 2023-07-07

**Soundness:** 3 good
**Presentation:** 2 fair
**Contribution:** 3 good
**Rating:** 5
**Confidence:** 4

**Summary:**

This paper introduces DropPos, a self-supervised pretext task designed to enhance the location awareness of Vision Transformers (ViTs). By dropping positional embeddings and reconstructing the positions of visible patches with some auxiliary strategies, DropPos improves spatial reasoning abilities in ViTs. Experimental results demonstrate the efficacy of DropPos, outperforming supervised pre-training and achieving competitive performance against state-of-the-art self-supervised methods. The paper provides good insights on enhancing location awareness in ViTs for future works.

**Strengths:**

1.	The motivation is clear. The paper addresses the motivation of enhancing positional awareness and spatial reasoning abilities in vision transformers for pre-training.

2.	The method is simple. The DropPos focuses on reconstructing dropped positions with some heuristics to avoid trivial solutions and ambiguities. Compare with the contrastive learning, it does not need complicated augmentations.

3.	The method is effective. Compared to contrastive learning or masked image modeling, the proposed DropPos exhibits faster pretraining and slightly improved performance on several benchmarks.


**Weaknesses:**

1.	The initialization of the positional encoding for the DropPos is not discussed, and there is no analysis of the impact of the different PE initialization strategies for the proposed DropPos method.
2.	The experiments are insufficient to validate the motivation. Although preliminary evidence is provided by experiments on downstream tasks such as detection and segmentation, which indicates that the DropPos enhances the location-awareness of ViT, the paper lacks more in-depth and intuitive experimental analysis and discussion to verify the strengthening of position sensitivity of ViT by the DropPos method.
3.	The description of the DropPos method is not specific and clear enough, including the method implementation, the flowchat of pseudo-code in section 3.2.
4.	The scaling properties of the DropPos on ViT are not explored in depth. For example, compared with the MAE method, It is questionable whether the method remains effective on Vit-Huge compared to the MAE. Experiments on more advanced vision transformers, such as Swin-Transformer, are also encouraged to be conducted.
5.	Compared with the MAE method, there is a lack of studies on transfer learning using iNaturalists and Places, and the robustness evaluation on ImageNet.


**Questions:**

1.	Regarding the DropPos method: Could you provide more specific details and clarity on the implementation of the DropPos method and the flowchart of the pseudo-code in Section 3.2? This would help readers better understand the proposed approach.
2.	Strengthening of the validation of motivation. Could you enhance the experimental analysis and discussion to provide more depth and intuition regarding the impact of DropPos on the position sensitivity of ViT? Can you consider conducting additional analyses or visualizations to better illustrate and explain the observed effects?
3.	Positional Encoding Initialization: It would be valuable to discuss the initialization strategy for positional encoding in the DropPos method. How was it initialized, and were different initialization strategies explored? Analyzing the impact of different positional encoding initialization strategies on the performance of DropPos would enhance the comprehensiveness of the study.
4.	Scaling Properties and Generalization: Could you further investigate and discuss the scaling properties of the DropPos method on larger architectures, such as ViT-Huge? Is the method equally effective and robust on larger-scale models? Additionally, have you considered applying the DropPos method to other advanced vision transformers like Swin-Transformer and evaluating its performance?
5.	Transfer Learning and Robustness Evaluation: It is noted that there is a lack of studies on transfer learning using iNaturalists and Places datasets, as well as the robustness evaluation on ImageNet. I would like to see if the DropPos method remains effective on these benchmarks.
6.	The Fig.~3 about the qualitative results of position reconstruction is somewhat confusing and requires a clearer explanation.

---

> ### Author Rebuttal · Authors · 2023-08-10
>
> We thank reviewer e3Vr for the valuable time and constructive feedback. Point-to-point responses
> are provided below.
>
> **Q1: The initialization of the positional encoding.**
>
> **A1:** DropPos uses fixed 2D sin-cos position embeddings by default. We ablate the initialization of
> position embeddings in the following table and it demonstrates that fixed sin-cos position embeddings
> achieves the best performance. We will add this in our revised version.
>
> | Initialization | Learnable | ImageNet | ADE20k |
> |---|---|---|---|
> | sin-cos | × | 82.96 | 40.68 |
> | sin-cos | √ | 82.81 | 39.37 |
> | random | √ | 82.48 | 38.72 |
>
> **Q2: The strengthening of improved position sensitivity.**
>
> **A2:** To systematically answer this question, we propose a metric to evaluate the model’s position
> sensitivity and explore the relationship between position sensitivity and performance on downstream
> tasks. Specifically, we freeze the backbone and train an extra linear position prediction head. Vanilla
> cross-entropy loss is used for training. Top-1 accuracies of position predictions before and after
> fine-tuning are reported, and 75% of position embeddings are randomly masked during training.
> Higher values mean the model is better at modeling the position relationship. The top-1 accuracy on
> the ImageNet validation set after fine-tuning is also reported.
>
> Please refer to Table 2 in the "global" response for detailed results.
>
> As shown in the table, the backbone performs better in position prediction after fine-tuning, indicating
> that *image classification indeed needs strong abilities in modeling spatial relationships*. It means that
> *better position sensitivity corresponds to better performances on downstream tasks*. By designing a
> position prediction pretext task, the backbone pre-trained by DropPos has better position modeling
> abilities, performing better on a variety of downstream tasks.
>
> **Q3: The implementation of DropPos is not specific and clear enough.**
>
> **A3:** We apologize for the ambiguity. To clarify the flowchart of DropPos, we provide the pseudo-code
> for computing the objective of DropPos. Please refer to the PDF in the "global" response for details.
> We will add this and polish the method section to make it clearer and more readable.
>
> **Q4: The scaling properties and generalization of DropPos.**
>
> **A4:** The scaling property is worth studying when evaluating the effectiveness of a self-supervised
> algorithm. Comparing performances between ViT-B and ViT-L, we can conclude that as the number
> of parameters in the model increases, the performance of DropPos is improving. However, the
> ViT-Huge training cannot be finished in the short rebuttal period, we will try to provide the results in
> the future revision.
>
> To verify the scaling property and the generalization of DropPos, we provide experiments when
> DropPos is equipped with the Swin Transformer. We follow the implementation of UM-MAE [a] and
> pre-train a Swin-Tiny from scratch using DropPos. All models are pre-trained with 200 epochs and
> fine-tuned with 100 epochs, following the configuration of UM-MAE [a].
>
> Please refer to Table 1 in the "global" response for detailed results.
> From the table, we can conclude that DropPos still works on Swin Transformers, and thus enhancing
> the location awareness of vision transformers is worth studying.
>
> **Q5: Lack of transfer learning results and the robustness evaluation.**
> **A5:** We recognize that these results are important to verify the generalization of a self-supervised
> algorithm. However, MAE evaluated the performance on iNaturalists and Places365 by fine-tuning
> on target datasets and the fine-tuning code has not been made publicly available. We will try to add
> these experiments in future revisions.
>
> We conduct robustness evaluation on ImageNet-Adversarial [b] and ImageNet-Rendition [c] using the
> same models fine-tuned on the original ImageNet and only run inference on the different validation
> sets in the following table, which is exactly the same as MAE. As shown in the table, with only 800
> epochs of pre-training, DropPos achieves comparable or even better performances, demonstrating its
> robustness.
>
> | Method | Backbone | Epoch | ImageNet-A [b] | ImageNet-R [c] | ImageNet |
> |---|---|---|---|---|---|
> | MAE | ViT-B | 1600 | **35.9** | 48.3 | 83.6 |
> | DropPos | ViT-B | 800 | 35.5 | **48.8** | **84.2** |
> | MAE | ViT-L | 1600 | **57.1** | **59.9** | **85.9** |
> | DropPos | ViT-L | 800 | 56.7 | 59.8 | 85.8 |
>
> **Q6: Figure 3 requires a clearer explanation.**
>
> **A6:** We apologize for the ambiguity. Figure 3 shows the qualitative results of position reconstruction
> under different mask ratios γ. Black patches are masked during inference. The positions of those
> white patches are wrongly predicted, while the remaining patches are predicted correctly. From the
> figure, we can conclude that DropPos manages to reconstruct precise positions even under extremely
> difficult situations (e.g., γ = 0.75).
>
> **References**
>
> [a] Xiang Li et al. Uniform Masking: Enabling MAE Pre-training for Pyramid-based Vision
> Transformers with Locality. arXiv:2205.10063, 2022.
>
> [b] Dan Hendrycks et al. Natural adversarial examples. In CVPR, 2021.
>
> [c] Dan Hendrycks et al. The many faces of robustness: A critical analysis of out-of-distribution
> generalization. In ICCV, 2021.

---

### Official Review · Reviewer_Pgbn · 2023-07-11

**Soundness:** 3 good
**Presentation:** 3 good
**Contribution:** 3 good
**Rating:** 6
**Confidence:** 5

**Summary:**

This paper a method for self-supervised representation learning. Given a ViT architecture,  the authors propose to predict the  absolute position of masked positional embedings at random. Although the general direction is not new, the authors pose it in a simple an interesting way, that achieves good performance in the downstream tasks.

**Strengths:**

1) The proposed ides is simple yet effective, and is executed well.
2) The presented is presented properly and is easy to follow.
3) The authors evaluate their method properly on several downstream tasks, achieving acceptable performance.

**Weaknesses:**

The direction of obtaining a supervision signal from the patch positions in the ViT architecture has been explored before. The authors mention the majority of them in their related work section but do not discuss what the advantage of their proposed method is. The performance supports the effectiveness of their method, but they could discuss more explicitly their advantages and ablate on that. Moreover, a comparison with [1] would be informative.

[1] Sameni et al., Representation Learning by Detecting Incorrect Location Embeddings, In AAAI 2022.

**Questions:**

Please see above.

**Limitations:**

Yes, they have.

---

> ### Author Rebuttal · Authors · 2023-08-09
>
> We thank reviewer Pgbn for the valuable time and constructive feedback. Point-to-point responses
> are provided below.
>
> **Q1: The advantage of the proposed DropPos should be discussed explicitly and lack of
> comparison with [a].**
>
> **A1:** Appreciate! We would like to discuss the advantage of our DropPos explicitly. As mentioned in
> the manuscript, there are three main difficulties in designing position-related self-supervised pretext
> tasks for ViTs. Specifically, (1) eliminating the Discrepancies between pre-training and fine-tuning,
> (2) avoiding trivial solutions to ensure highly semantic representations, and (3) reducing the impact of
> confusing targets caused by similar visual appearances. All other methods cannot manage all of them,
> which is summarized in the following table.
>
> | Method | Eliminate Discrepancies | Avoid Trivial Solutions | Remove Confusing targets |
> |-------------------|-------------------------|-------------------------|--------------------------|
> | Zhai et al. [60]  | × | × | × |
> | Caron et al. [4]  | √ | × | √ |
> | Sameni et al. [a] | √ | × | √ |
> | DropPos           | √ | √ | √ |
>
>
> Specifically, Zhai et al. [60] simply discard all positional embeddings during pre-training, and
> thus discrepancies arise. Therefore, the fine-tuning performances of [60] largely lag behind the
> state-of-the-art.
>
> Caron et al. [4] propose to predict the relative location of a local crop to the corresponding global
> crop, making it time-consuming and hard to learn highly semantic representations as this pretext task
> is somewhat too simple for powerful ViTs. ViTs may simply solve this task by comparing the texture
> of two given crops.
>
> Sameni et al. [a] come up with an auxiliary position-related objective and combine it with the popular
> contrastive learning paradigm. Therefore, the generalization abilities of learned representations
> are highly related to data augmentation techniques. Also, the position-related task itself proposed
> by [a] (without contrastive learning) may become a trivial solution, as identifying several mismatched
> positions is relatively easy for powerful ViTs.
>
> DropPos solves the mentioned three difficulties by dropping a subset of position embeddings, dropping
> a large subset of patch tokens, and position smoothing and attentive reconstruction, respectively.
> Despite these advantages, the pre-training procedure of DropPos is ≈ 3× more efficient than [60, 4,
> a] thanks to the patch masking stage.
>
> In fact, all these three advantages have been ablated in the manuscript. Specifically, we ablated
> the effectiveness of alleviating discrepancies in Table 2 (γ_pos = 0.75 *v.s.* γ_pos = 1), where an
> improvement of +0.3% fine-tuning top-1 accuracy is observed. The second advantage is ablated
> in Table 1 (γ = 0.75 *v.s.* γ = 0), where an improvement of +1.02% fine-tuning top-1 accuracy is
> observed. The effectiveness of eliminating confusing reconstruction targets is ablated in Table 3
> (σ = 1 → 0 *v.s.* σ = 0) and Table 4 (τ = 0.1 *v.s.* τ = ∞), where improvements of +0.11% and
> +0.12% fine-tuning top-1 accuracies are observed.
>
> We will emphasize these results in the revision to further help address the concerns.
>
> **References**
>
> [a] Sepehr Sameni et al. Representation Learning by Detecting Incorrect Location Embeddings. In
> AAAI 2022.

---

### Author Rebuttal · Authors · 2023-08-09

To all reviewers:

Thank you so much for your careful review and suggestive comments. Following your suggestions, we present some extra figures and tables in the PDF. We also provide the pseudo-code for computing the objective of DropPos. Specifically,
- **@Reviewer e3Vr**, to clarify the flowchart of DropPos, we provide the pseudo-code for computing the objective in Algorithm 1.
- **@Reviewer 6DGy**, to evaluate the sparsity of attention maps of DropPos, we compare attention maps generated by MAE and DropPos at the first and the last Transformer blocks in Figure 1.
- **@Reviewer 6DGy**, to measure the receptive field of DropPos, we compare the mean attention distance of MAE and DropPos in Figure 2.
- **@Reviewer e3Vr** and **Reviewer 6DGy**, we conduct experiments in Table 1 to verify that DropPos also works with Swin Transformers.
- **@Reviewer e3Vr**, **Reviewer 6DGy**, and **S2Rp**, to explore the strengthening of improved location awareness, we provide an in-depth and intuitive experimental analysis in Table 2, where we freeze the pre-trained backbone and train an extra linear patch classification head.

Sincerely,

Authors.

---

### Decision · Program_Chairs · 2023-09-21

**Decision:**

Accept (poster)

**Comment:**

All the five reviewers recommend acceptance. The authors should add the clarifications in their rebuttal to the final version.